# Structure of Alloys for (Sm,Zr)(Co,Cu,Fe)_Z_ Permanent Magnets: First Level of Heterogeneity

**DOI:** 10.3390/ma13173893

**Published:** 2020-09-03

**Authors:** Andrey G. Dormidontov, Natalia B. Kolchugina, Nikolay A. Dormidontov, Yury V. Milov

**Affiliations:** LLC “MEM”, Moscow 123458, Russia; natalik014@yandex.ru (N.B.K.); ontip@mail.ru (N.A.D.); milov.yv@mail.ru (Y.V.M.)

**Keywords:** Sm–Co permanent magnet alloys, highly coercive structures, composition-structure-properties diagrams, (Co,Cu,Fe)–R–Zr phase diagrams sketches

## Abstract

An original vision for the structural formation of (Sm,Zr)(Co,Cu,Fe)_Z_ alloys, the compositions of which show promise for manufacturing high-coercivity permanent magnets, is reported. Foundations arising from the quantitative analysis of alloy microstructures as the first, coarse, level of heterogeneity are considered. The structure of the alloys, in optical resolutions, is shown to be characterized by three structural phase components, which are denoted as A, B, and C and based on the 1:5, 2:17, and 2:7 phases, respectively. As the chemical composition of alloys changes monotonically, the quantitative relationships of the components A, B, and C vary over wide ranges. In this case, the hysteretic properties of the (Sm,Zr)(Co,Cu,Fe)_Z_ alloys in the high-coercivity state are strictly controlled by the volume fractions of the A and B structural components. Based on quantitative relationships of the A, B, and C structural components for the (R,Zr)(Co,Cu,Fe)_Z_ alloys with R = Gd or Sm, sketches of quasi-ternary sections of the (Co,Cu,Fe)-R-Zr phase diagrams at temperatures of 1160–1190 °C and isopleths for the 2:17–2:7 phase composition range of the (Co,Cu,Fe)–Sm–Zr system were constructed.

## 1. Introduction

Permanent magnets based on the (Sm,Zr)(Co,Cu,Fe)_Z_ alloys, which were developed and commercialized a while ago [1,2], are complex metallurgical systems and their hysteretic properties and structural state are closely linked to the chemical composition of the material and heat treatment conditions. Despite numerous investigations into the structural formation and properties of the alloys, the studies are far from over. The phase transformations, which occur during stepped tempering, and the transformations that cause the instant “recovery of properties” upon repeated heating of samples to the isothermal tempering temperature being among them [3,4], have not yet been described. The phase composition of boundaries between cells at different states of heat treatments [3,4,5,6] meets with strongly conflicting views. There are sufficiently exotic variants of the phase structure at the boundary–cell interface [7]. For example, Popov et al. [7] assume that the boundary phase is separated so that Cu is localized in the 1:5 phase near the (1:5)/(2:17) interface rather than in the center of the 1:5 boundary phase. However, the structural peculiarity of the boundary is the alternation of phase layers strictly across the “*c*” axis, which is common for the whole anisotropic massive of samples [3,8]. Within such a structure, the formation of additional phase separation along the most low-size coordinate of boundary structural element is not likely energetically reasonable. Moreover, this assumption certainly contradicts with the fact that the Cu concentration in the center of the boundary phase in ternary junctions of the cells is maximal [8].

It should be recognized that currently there is no objective holistic view of the formation of high-coercivity structure of the (Sm,Zr)(Co,Cu,Fe)_Z_ alloys. This hinders the progress in the development and improvement of both new compositions of alloys for permanent magnets and manufacturing processes for them.

The aim of the present study was to detail the interrelations of the chemical and phase compositions of the (Sm,Zr)(Co,Cu,Fe)_Z_ alloys in the composition ranges, which are of importance for the manufacturing of permanent magnets because of the functionality of their phase structure. The magnetic hardness of the (Sm,Zr)(Co,Cu,Fe)_Z_ alloy for permanent magnets is ensured by its precipitation hardening in the course of complex heat treatment. The detailing is performed based on the generalizing of both original results, which are not given in widely available periodicals and the known literature data. The study is also aimed at concretizing the phase transformations of the Sm–Zr–Co–Cu–Fe system in accordance with our understanding of the occurred processes and phenomena.

## 2. Methodological Features

### 2.1. On the Formula of Alloys for the (Sm,Zr)(Co,Cu,Fe)_Z_ Permanent Magnets

Among the fundamental studies that have determined the methodology of investigations of (Sm,Zr)(Co,Cu,Fe)_Z_ alloys, there are investigations related to Ray’s metallurgical model [9,10] in which the process of the formation of the high-coercivity structure was improved. The behavior of alloys during heat treatments can be determined by the following three main model postulates:The starting point is the matrix, which is the disordered single-phase 2:17R precursor; the decomposition of the phase during isothermal tempering leads to the formation of the cellular-lamellar structure of phases. The formed phases are modified at the expense of interphase interdiffusion during stepped aging.Zirconium with a vacancy occupies Co_2_ dumbbell sites when entering the Sm_2_(Co,Fe,Cu,Zr)_17_ matrix.The composition of the alloy for manufacturing the permanent magnets should be maximally close to the 2:17 stoichiometry taking into account Sm and Zr losses for the oxidation and reactions with other nonmetallic impurities.

In this context, the formulas Sm_2_(Co,Fe,Cu,Zr)_17_ and Sm(Co,Cu,Fe,Zr)_Z_ have been logically and widely used by investigators not only for the generic designation of alloys and magnets based on them, but also for plotted dependences “composition–physical properties”.

We assumed that the alloy formula of Sm_1-X_Zr_X_(Co,Cu,Fe)_Z_ suggested by researchers of the Department of Magnetism, Tver State University supervised by Prof. Dmitry D. Mishin, Ph.D (M.B. Lyakhova et al. [11,12]) is physically more true. In the case of such a formula, at least, the dependences composition-properties can be concretized personally for 4f, 4d, and 3d elements of the alloy. There are additional arguments in favor of the formula.

Currently, it is obvious that during heat treatment for the high-coercivity state, at the end of the solid solution heat treatment (SSHT), the starting matrix, in the classical sense of this term, is formed based on the disordered 1:7H phase [3,4,5,6] (from herein, the designation of A_n_B_m_ intermetallics is given in the form of n:mH or n:mR, where H and R indicate the hexagonal and rhombohedral crystal lattices, respectively).

The 1:7H structure is the hexagonal TbCu_7_-type phase (space group P6/mmm), rather than 2:17R, which is typical of 1:5H and 2:17H [13] or P6_3_/mmc [14]. According to Lefevre et al. [14], the hexagonal structure is not a disordered form of the rhombohedral Sm_2_Co_17_ phase. It is obtained by deformation of the SmCo_5_ structure through the disordered SmCo_7_ structure. In such a structure, Zr atoms occupy Sm sites.

The substitution of Zr atoms in the close hexagonal Th_2_Ni_17_-type structure refined by the Rietveld method corresponds approximately to 0.5/0.5, 0.75/0.25, and 0.9/0.1 Sm/Zr or Co_2_/Zr population on sites 2b, 2d, and 4f/2c, respectively. Thus, the preferred sites for accepted Zr atoms are the Sm atom sites [14].

The analogous situation is characteristic for the Fe–Gd–Zr system. The Rietveld refinement of the Fe_17_(Gd,Zr)_2_ structure indicated that Zr atoms reside only on the 2b site [15,16].

Thus, the matrix being the initial solid solution, with good reason, is written in the form of Sm_1-X_Zr_X_(Co,Cu,Fe)_Z_.

The substitutions Sm–Zr in crystal lattices of phases of the Sm_n-1_Co_5n-1_ homological series has not been in doubt for a long time. As for the 2:17R phase, Zr should definitely be included in parentheses together with the Co or 3d-metal mixture, Sm_2_(Co,Zr)_17_. However, this phase, despite its high final volume in the alloy and high role in the formation of magnetic properties, is the secondary product of phase transformations.

Thus, the arguments for the original formula of the Sm_1-X_Zr_X_(Co,Cu,Fe)_Z_ alloys are:-the separation of the effect of main elements qualitatively differing in the configurations of electron shells (4f–Sm, 4d–Zr, and 3d–Co,Cu,Fe) on the structure and properties of the studied alloys; and-during the formation of the alloy structure in the course of heat treatment, the matrix (the initial structural state of test object) is the disordered 1:7H solid solution and almost all intermediate and final phases formed in the course of phase transformations, except 2:17R, adequately correspond to the formula Sm_1-X_Zr_X_(Co,Cu,Fe)_Z_.

### 2.2. On the Heterogeneity of Alloys for the (Sm,Zr)(Co,Cu,Fe)z Permanent Magnets

There are many results of experimental investigations of (Sm,Zr)(Co,Cu,Fe)_z_ materials for permanent magnets in the literature, which were obtained by modern experimental equipment that is perfect from the point of view of the methodological approach [3,5,6,7,8].

Results obtained by methods including high-resolution transmission electron microscopy (HRTEM), nanoprobe energy dispersive x-ray spectroscopy (EDXS), nano-beam diffraction (NBD), etc. have allowed one to estimate the fine structure of (Sm,Zr)(Co,Cu,Fe)_Z_ sintered magnet samples using results of the local analysis of some phases [3,5,6,8]. However, sometimes, the relation of results obtained for the structure of magnets and the initial alloy structure is overlooked. It is obvious that the real sintered rare-earth permanent magnets are produced from thin powders, otherwise, the energy potential cannot be realized completely. It should be noted that the homogenization of the alloy in the course of magnet production starts directly after milling rather than in reaching the working temperature of SSHT. The manufacturing operations (milling, sintering, and SSHT) homogenizing the alloy shift the experimental pattern of phase transformations from realism to impressionism, which makes their interpretation difficult.

The methodology of this study was built taking into account the evidence of several levels of heterogeneity of the (Sm,Zr)(Co,Cu,Fe)_Z_ alloys and magnets based on them:The first obvious level of heterogeneity of the (Sm,Zr)(Co,Cu,Fe)_Z_ alloys, which was noted in numerous studies, can be observed with optical resolutions [11,12,17,18,19,20,21]. It is characterized by three clear components comprising the structure, which, with optical magnifications, have signs of all phases, which, with the higher magnifications, are obviously heterogeneous. We denoted these “optical phases” as structural phase components (SPC).The second level of heterogeneity typical of both (Sm,Zr)(Co,Cu,Fe)_Z_ alloys and magnets based on them is the scale of the so-called cellular structure, which has been repeatedly studied, and consist of cells and cell boundaries that are penetrated by thin plates along the basal planes.The third level of heterogeneity is the structure of boundaries between cells, which obviously undergo phase transformations at all stages of thermal aging of the (Sm,Zr)(Co,Cu,Fe)_Z_ alloys and magnets based on them.

This study was devoted to regularities mainly typical of the first, the most rough, level of the heterogeneity of materials based on the (Sm,Zr)(Co,Cu,Fe)_Z_ alloys.

## 3. Materials and Methods 

The (Sm,Zr)(Co,Cu,Fe)_Z_ alloys were prepared by high-frequency induction melting in an argon atmosphere at an excess pressure of 5–10% using Al_2_O_3_ crucibles (NTC Bakor, Ltd, Moscow, Russia) and individual components such as Sm, Zr, Co, Fe, and Cu of >99.9, >99.97, >99.98, >99.7, and >99.97 wt.% purity, respectively. After melting, the alloys were not cast in a mold but slowly cooled in the crucible, which was carefully thermally isolated from a water-cooled inductor. As a result, an ingot characterized by coarse grains 4–6 mm in average size was obtained. With respect to the magnetic state, each grain had a single easy magnetization axis (EMA). These samples could not be classified among traditional single crystals since they are not single-phase. However, all phases of one grain were strictly collinear; in other words, the EMAs of each phase within a grain were parallel to each other. For simplicity, from here on, we call these samples pseudo-single crystals.

The chemical composition of the Sm_1-X_Zr_X_(Co_1-a-b_Cu_a_Fe_b_)_Z_ samples of the experimental series is given in Table 1.

The chemical analysis of alloys was performed by optical emission spectroscopy with inductively coupled plasma (ULTIMA 2 Jobin-yvon ICP-OES, Tokyo, Japan).

The heat treatment conditions used were similar to those described in the literature [2,3,4,5,6,7,8]. Heat treatment for the solid solution (SSHT) at 1150–1180 °C for 5 h and subsequent water cooling were performed. The isothermal tempering at 800 °C was carried out for 20 h; after that, samples were either water-quenched or subjected to step tempering to 400 °C at an average cooling rate of 100°/h followed by furnace cooling. All heat treatments were performed after preliminary degassing during heating to 600 °C in a vacuum and subsequent high-purity argon puffing to a pressure that slightly exceeded (to 5–10%) the atmospheric pressure.

Pseudo-single crystal samples of each of the compositions in as-cast and heat-treated (under different conditions) states were ground to form balls 2.5–3.5 mm in diameter by grinding with gaseous nitrogen supplied under a pressure between two abrasive caps.

The magnetic measurements of major and minor hysteresis loops of pseudo-single crystal spherical samples were performed at room temperature using a vibrating sample magnetometer (VSM LDJ Electronics Inc., Model 9600, Troy, NY, USA) (H_MAX_ = 30 kOe). To measure the major hysteresis loop, the samples were preliminarily magnetized in a magnetic field (Ningbo Canmag Electronics Co., Model KCJ-3560G, Ningbo, China) of no less than 100 kOe. The VSM was graduated using a Ni standard with σ_SNi_ = 54.4 G × cm^3^/g. A VSM-option of PPMS-9T Quantum Design installation was used to study the magnetic characteristics of samples, the coercive force of which at room temperature is above 30 kOe.

The samples completely evaluated with respect to magnetic parameters were demagnetized using alternating magnetic field of VSM with decreasing amplitude and were fixed in mandrels with a fast curing epoxy compound.

The fixation of samples was performed using a magnetic field and a specific facility in order to align the basal or prismatic planes of the sample in parallel to the surface of the prepared section. The sections were prepared in accordance with standard techniques using diamond pastes with a gradual reduction in grain size. The mandrels for the fixations of samples allowed us to mount them in a holder of VSM and perform additional manipulations in the applied magnetic field.

The polished and etched sections were studied using optical magnifications in order to visualize the domain structure by Kerr-effect method and the microstructure, respectively. The quantitative relationships of structural components were determined in accordance with standard stereometric metallography techniques.

The fine structure of samples was studied by scanning electron microscopy (SEM) (using a scanning electron microscope (FEI Company, Fremont, CA, USA with EDS, EDAX Inc. Mahwah, NJ, USA). The compositions of structural elements were studied by local x-ray energy dispersive analysis using pure elements as the standards. The EDX analysis of each of samples started from the determination of the integral chemical composition for the maximally possible section area in order to control the correspondence of the composition of the sample to the chemical analysis data for the associated series. Analogously, individual structural components were studied using areas knowingly apart from the interphase boundaries. Element scanning across the boundaries of the structural components was also used.

Anisotropic sintered powder samples were prepared from these alloys using a standard powder metallurgy technique, which includes crashing in a nitrogen atmosphere, and milling in a vibrating ball mill. Compaction of powders in a transverse magnetic field, cold isostatic pressing, sintering in a hydrogen atmosphere, and heat treatment in an argon atmosphere were performed in accordance with regimes described above. The magnetic properties of the sintered samples (Ø15 × 7 mm) were measured in fields of 30 kOe using a completely closed magnetic circuit and an automatic recording flux meter (B-H tracer LDJ Electronics Inc., Model 5500H, Troy, NY, USA).

## 4. Results

### 4.1. Dependence of Magnetic Properties of Samples on the Chemical Composition of the (Sm,Zr)(Co,Cu,Fe)_z_ Alloys

Figure 1 shows the major and minor hysteresis loops of the Sm_0.87_Zr_0.13_(Co_0.690_Cu_0.070_Fe_0.240_)_6.5_ sample, which are typical of all samples under study.

The given hysteresis loop is sufficiently wide for detailing the minor hysteresis loops and is also almost completely within the operating fields of VSM.

The bottom curves correspond to the magnetization curves of sample demagnetized by reversing field and the series of minor hysteresis loops measured with progressively increased magnetized field. The top curves indicate magnetization curves of the same sample demagnetized with the alternating field with decreasing amplitude and the series of minor hysteresis loops measured with progressively increased magnetized field.

This sample and samples of all alloys under study indicate the classic magnetic hysteresis related to the domain wall pinning mechanism.

It can be seen that from the viewpoint of maximum energy product, the hysteresis loop is ultimate. The decrease in the magnetization, with allowance for the demagnetizing factor, starts in negative fields of ~8–9 kOe; the specific saturation magnetization of the sample was σ_S_ = 116 G × cm^3^/g, which corresponds to values 4πJ_S_ = 12.5 kG and (BH)_MAX_ = 39 MGOe.

The hysteresis loops of the other samples are qualitatively similar to that given in Figure 1. The differences consist in scales of loops. Slight differences were observed along the ordinate axis, which were in accordance with the relative concentrations of Co and Fe, and significant differences were observed along the abscissa axis, which were due to the structural compositions of the samples.

The magnetization reversal portions of the major hysteresis loops of samples of series no. 2 (Table 1) with the fixed x, Sm_0.85_Zr_0.15_(Co_0.702_Cu_0.088_Fe_0.210_)_Z_, and samples of series nos. 1–4 with the fixed z, Sm_1-X_Zr_X_(Co_0.702_Cu_0.088_Fe_0.210_)_6.4_, are given in Figure 2.

It can be seen clearly that as z increased for the Sm_0.85_Zr_0.15_(Co_0.702_Cu_0.088_Fe_0.210_)_z_ alloys (Figure 2a), the coercive force increased monotonously over the whole composition range. However, beginning from certain z, at low values of applied magnetic field, the magnetization reversal curve indicated a “shoulder” (i.e., the squareness of the loop decreases). Such behavior is completely typical of all samples under study. 

The difference consists only in the value of the “shoulder”. The lower the relative zirconium concentration (x), the clearer the “shoulder”. Moreover, as the relative zirconium concentration (x) increases, the z value corresponding to the worsening loop squareness shifts to the low values. This can be clearly observed in comparing the dependences in Figure 2b.

The dependences of the magnetic properties of the sintered powder samples on the chemical composition of the Sm_1-X_Zr_X_(Co,Cu,Fe)_z_ alloys were qualitatively similar to those of the pseudo-single crystal samples. As an example, Figure 3 shows the characteristic magnetization reversal curves of sintered powder samples in the high-coercivity state (series no. 2).

The main difference of the sintered powder samples consists of the fact that the characteristic “shoulder” in the magnetization reversal curves of powder samples appeared for the compositions with z lower by 0.15–0.2 than that for the pseudo-single crystal samples of analogous compositions from all series of alloys under study. It is likely that the decrease in the threshold of the appearance of the “shoulder” is related to samarium and zirconium losses due to the oxidation and to the reactions with nonmetallic impurities and samarium losses due to evaporation during the course of manufacturing the sintered samples.

### 4.2. Dependence of the Structure of Samples on the Chemical Composition of the (Sm,Zr)(Co,Cu,Fe)_z_ Alloys

Figure 4 shows the microstructure on the prismatic plane of the pseudo-single crystal sample of Sm_0.85_Zr_0.15_(Co_0.702_Cu_0.088_Fe_0.210_)_6.4_ alloy in the (a) as-cast state, (b) after solid solution heat treatment, and (c) after complete cycle of heat treatment for the high-coercivity state, which are typical of all series of samples.

The microstructure of all samples under study consisted of three structural phase components (SPCs): dark-grey SPC (from here on, A); bright-grey dendritic-like SPC (B), and almost white anisotropic SPC elongated along the basal plane of sample (C), which, for all samples, was localized within SPC A. 

The heat treatment did not lead to any substantive changes in the quantitative relations of structural components in all studied samples with the same chemical composition. After solid solution heat treatment and quenching, the characteristic composition contrast in the section disappeared; however, the cast prehistory of the cast state was clearly observed. After a complete heat treatment cycle, the contrast recovered but differed in the halftone pattern.

A portion of the microstructure in Figure 4c was contrasted in order to perform the quantitative analysis of the image. 

The same colors were used in Figure 5 to show changes in the relationships of structural components with progressive change in the chemical composition of the alloys of series nos. 1, 2, and 4 (Table 1). It is clearly observed that the microstructures of all studied samples in the high-coercivity state were qualitatively identical and were the superposition of three structural components A, B, and C, and the quantitative relationships between which varied monotonously with changing chemical composition within each series of alloys. 

The energy-dispersive X-ray (EDX) microanalysis performed for the samples of all series of alloys in the high-coercivity state showed the similarity of all chemical compositions of the structural components. It is possible to state that the intermediate compositions corresponded to the common formulas: in SPC A, (Sm,Zr)(Co,Cu,Fe)_6.3_; in SPC B, (Sm,Zr)(Co,Cu,Fe)_7.1_; and in SPC C, (Sm,Zr)(Co,Cu,Fe)_3.7_ (in alternative variants, Sm(Co,Cu,Fe,Zr)_7.5_, Sm(Co,Cu,Fe,Zr)_8.4_, and Sm(Co,Cu,Fe,Zr)_12_, respectively). Thus, the structural components were formed based on the 1:5 (A), 2:17 (B), and 2:7 (C) phases. This corresponds to results reported by Kianvash et al. [17].

### 4.3. Interrelations between the Chemical Composition, Structure, and Hysteresis Parameters of (Sm,Zr)(Co,Cu,Fe)_z_ Alloys in the High-Coercivity State

Figure 6 shows the dependences of the coercive force and quantitative relationships of the structural components on the chemical composition of samples in the high-coercivity state for the Sm_1-X_Zr_X_(Co_0.702_Cu_0.088_Fe_0.210_)_z_ alloys with the same relation of 3d elements (series 1–4).

The analysis of the dependences given in Figure 6 allowed us to infer the following:

As the relative Zr content (x) increases, the volume percentage of the structural component C increases. In this case, for each of the series of alloys, as the content of 3d elements (z) increased, the volume percentage of structural component C decreased.

The volume percentages of structural components A and B corresponding to more than 90% of the total alloy volume changed monotonously (for all experimental series) from the dominating volume percentage of structural component A to the dominating volume percentage of structural component B.

As the relative content of 3d elements (z) and the volume percentage of structural component B increased in compositions corresponding to the equality of the volumes of structural components A and B (V_A_ = V_B_), an intense increase in the coercive force was observed; after that (at V_A_ < V_B_), the increase in H_CJ_ decelerated and the dependence reached the horizontal “plateau”.

The comparison of the data given in Figure 2a,b and Figure 5b shows that for the series of Sm_1-X_Zr_X_(Co_0.702_Cu_0.088_Fe_0.210_)_Z_ alloys, the transition through the chemical composition corresponding to the equality of volume contents of structural components A and B (V_A_ = V_B_) was accompanied by the appearance of a “shoulder” in the magnetization reversal curves along with the end to increase in H_CJ_. Thus, the reaching of the alloy chemical compositions corresponding to the condition V_B_ > V_A_ was accompanied by the decrease in the squareness of hysteresis loops. This was satisfied for all series of Sm_1-X_Zr_X_(Co,Cu,Fe)_Z_ alloys.

Figure 7 also demonstrates the validity of the noted regularities for series no. 5, characterized by the other relationship of 3d elements in the Sm_0.85_Zr_0.15_(Co_0.665_Cu_0.075_Fe_0.260_)_Z_ alloy. Thus, the following conclusion can be inferred.

The major hysteresis loops for the Sm_1-X_Zr_X_(Co,Cu,Fe)_Z_ alloys in the high-coercivity state, which are characterized by the maximum (for a certain relation of alloy components) coercive force (H_CJ_) along with the high squareness of the loop, are reached in the case of equal volumes of structural components A and B comprising the main volume of the alloy and largely determining the hysteretic properties of the alloy.

This conclusion also fully corresponds to samples of sintered magnets produced from the powders of the alloys in the experimental series (with allowance for notes given above in the caption for Figure 3).

## 5. Discussion

### 5.1. Alloy-Formation in the Co–Sm–Zr and (Co,Cu,Fe)–Sm–Zr Systems

Note that it is difficult to compare the data on the phase diagrams constructed by different investigators, which use different coordinate systems. Due to this, we compiled the available data and constructed ternary phase diagrams in a form more convenient for the present study, namely, cobalt was always in the left corner of the binary, quasi-binary sections, and ternary phase diagrams; rare-earth metal was always in the top corner of the ternary systems. The alloying component was in the right corner.

The structural formation of the Sm–Zr–Co and Sm–Zr–Co–Cu–Fe alloys has been considered in a great number of studies [14,17,18,19,20,21,22,23,24,25,26,27,28,29,30,31]. The analysis of these results allowed us to formulate the following regulations.

The introduction of zirconium into the alloy leads to the formation of a wide TbCu_7_-based solid solution region in the composition range of existence in the 2:17 and 1:5 phases. The 1:7 region was typical of high temperatures corresponding to the SSHT temperatures of the (Sm,Zr)(Co,Cu,Fe)_Z_ alloys and is directly adjacent to the homogeneity range of the 2:17 phase in the 2:17H modification with the Th_2_Ni_17_-type structure (Figure 8 [25]).

The decomposition of the solid solution led to the precipitation of the 2:17 phase and continuous formation of phases of the homologous row Sm_n-1_Co_5n-1_.

The analysis of numerous microstructures of Sm_1-X_Zr_X_(Co,Cu,Fe)_Z_ samples under study in the as-cast and heat treated states shows that, for all of the studied compositions (Table 1) upon solidification and subsequent solid solution decomposition, the discontinuity was characteristic for two primary independently solidified 2:17 and 2:7 phases (Figure 4). The phase components B and C were formed based on the phases.

Regarding structural phase component A, its structure is likely to correspond to the statement [26,27] on continuous phase formation.

Features of the structural formation at the 2:17 phase side, which are given in Morita et al. [22,23,24], were collected by us as a logical sequence in Figure 9.

Note that when plotting the compiled data of Morita et al. [22,23,24], we allowed ourselves to correct some inherent original obvious annoying inaccuracies.

However, in our opinion, to completely present the alloy formation of the R_1-X_Zr_X_(Co,Cu,Fe)_Z_ compositions, the structural formation at the side of the 2:7 phase is also of importance. To refine the processes that occurred, we considered the previously published results.

### 5.2. Alloy-Formation in the (Co,Cu,Fe)–Gd–Zr Systems

For the first time, the existence of the strict correspondence between the alloy composition, alloy microstructure (in optical resolutions), and hysteretic properties, which is given in Section 4.2, was found by Lyakhova et al. [11,12] for the Gd_1-X_Zr_X_(Co,Cu,Fe)_Z_ compositions.

It should be noted that the Gd-containing alloys were found to be preferred as model alloys for the investigation of composition–property correlations. The ferrimagnetic coupling of rare-earth and transition metal sublattices of the main phases decreased the magnetic moment of the samples and led to lower anisotropy field and, therefore, coercive force values. This is convenient for magnetic measurements in the limited range of quasi-static external magnetic fields. Moreover, the oxidation of Gd-containing alloys is lower than that of Sm-containing alloys, and the vapor pressure of Gd at 1000 °C is lower than that of Sm by seven orders of magnitude. These circumstances facilitate the preparation of samples. The microstructure of the Gd_1-X_Zr_X_(Co,Cu,Fe)_Z_ alloys can be easily etched and are more in contrast when compared to that of the Sm alloys because of the more simple set and configuration of phases observed for the actual composition range. In particular, the 5:19 phase was absent in the Gd–Co system. In the rare-earth series, when the atomic number increased, the atomic radius decreased and the 2:17H structure was stabilized, in particular, when Zr atoms were substituted for R atoms [14].

It should be noted that Khan [32] had already noted a certain similarity of the Sm–Co and Gd–Co systems, namely, the existence of the 1:7 phase, as a polymorphic modification of the 2:17 phase existing at high temperatures without introducing an additional alloying element, only in binary cobalt systems with samarium and gadolinium.

Lyakhova et al. [11,12] showed the qualitative identity of the dependences of the volume of structural components and magnetic properties on the chemical composition over a wide composition range of Gd_1-X_Zr_X_(Co,Cu,Fe)_Z_ alloys (Figure 10).

Figure 10a shows the positions of the experimental series of Gd_1-X_Zr_X_(Co,Cu,Fe)_Z_ alloys in the (Co,Cu,Fe) corner of the quasi-ternary (Co,Cu,Fe)–Gd-Zr system (Figure 10b–h). The dependences properties vs. composition are shown by corresponding letters.

As a whole, the experimental dependences for the Gd_1-X_Zr_X_(Co,Cu,Fe)_Z_ alloys were similar to those for the Sm_1-X_Zr_X_(Co,Cu,Fe)_Z_ alloys (Figure 5, Figure 6 and Figure 7).

The difference between the data given for the (Gd,Zr)(Co,Cu,Fe)_Z_ alloys and those given for the (Sm,Zr)(Co,Cu,Fe)_Z_ alloys consists of the fact that because of the peculiarities noted for the Gd alloys, the dependence of the coercive force on the composition for all series exhibited a clear extremal behavior. The maximum of the dependence of coercive force strictly corresponds to the equality of the volumes of structural components A and B.

All the conclusions formulated by us based on the results obtained for the Sm-containing series of alloys were satisfied for the Gd-containing alloys.

However, the extremal behavior of the dependences of the coercive force on the chemical and phase compositions for the Gd alloys allowed us to made an additional conclusion formulated for the (Sm,Zr)(Co,Cu,Fe)_Z_ alloys.

For the experimental series of (Gd,Zr)(Co,Cu,Fe)_Z_ alloys, a trend was clearly observed to broaden the coercivity peak (decrease in sharpness) with increasing Zr content and trend to shifting the peak to the low relative fraction of 3d-elements (z).

The (Sm,Zr)(Co,Cu,Fe)_Z_ samples in the high-coercivity state (see Figure 1, Figure 2, and Figure 7) and pseudo-single crystals (Gd,Zr)(Co,Cu,Fe)_Z_ in the high-coercivity state demonstrate the ultimate hysteresis loops (Figure 11).

The comparison of Figure 10c and Figure 11 clearly shows that as the chemical composition changes monotonically in the Gd_0.85_Zr_0.15_(Co_0.70_Cu_0.09_Fe_0.21_)z alloy series, the excess of the volume content of component B over component A (V_B_ > V_A_) is accompanied by the decrease in the coercivity and, like in the case of (Sm,Zr)(Co,Cu,Fe)_Z_ alloys, by the progressive decrease in the squareness of the hysteresis loop.

We calculated the phase relations for the (Co,Cu,Fe)–Gd–Zr system based on results given in Figure 10 taking into account the behavior of the structure-sensitive characteristic such as the intrinsic coercive force of samples (H_CJ_), localization of Zr in crystal lattices of Sm_n-1_Co_5n-1_ phases, and the above constructed Co–Sm–Zr and (Co,Cu,Fe)–Sm–Zr phase diagrams.

The results are given in Figure 12.

This sketch of the diagram corresponds to temperatures slightly below the upper interval of SSHT temperature, at which, according to results of the analysis of microstructures of the (R,Zr)(Co,Cu,Fe)_z_ alloys, obvious melting on the edge of the structural component C (2:7) was observed. Similar melting was observed by Kianvash et al. [17].

### 5.3. Alloy-Formation in the (Co,Cu,Fe)–Sm–Zr Systems

Taking into account the data given in Figure 6, the phase relations in the Gd_1-X_Zr_X_(Co,Cu,Fe)_Z_ alloys (Figure 12) easily transform into the corresponding diagram for the Sm_1-X_Zr_X_(Co,Cu,Fe)_Z_ alloys.

Calculations show that the transition from the constructions performed for the Gd-containing compositions to the constructions for the Sm-containing compositions was mainly reached by elongation of the diagram in Figure 12 along the equi-composition axes R–Zr. In this case, it should be taken into account that the diagram for Sm alloys has an additional “arrow” of the 5:19 phase and areas of interaction of the phase with neighboring phases.

The obtained sketch of the diagram, which indicates the phase relations in the Sm_1-X_Zr_X_(Co,Cu,Fe)_Z_ alloys (used for manufacturing of permanent magnets) in the temperature range of SSHT, is given in Figure 13.

Figure 14 shows how, in accordance with our results, the binary Co–Sm phase diagram transforms in adding alloying components (Zr,Cu, and Fe) for the section shown by the dashed line in Figure 13.

As can be seen from Figure 12, Figure 13 and Figure 14, the phase formation in the five-component (R,Zr)(Co,Cu,Fe)_Z_ system differed substantially from the diagrams given in [25,26,27,28,29,30,31] and mainly, does not contradict the data of Morita et al. [22,23,24].

As the Zr content in the alloys increases, the region of the high-temperature 1:7 phase, on the way out the “corridor” of the wide homogeneity range of the derivative 1:(5 + x) phase, “runs” in two directions corresponding to the back course of branches of the primary solidification of the alloy from liquid.

The first is the direction toward the homogeneity range of the high-temperature hexagonal polymorphous modification of the 2:17 phase (space group *P6/mmm*). The second is the direction toward the homogeneity range of high-temperature hexagonal polymorphous modification of the 2:7 phase (space group *P6_3_/mmc*).

The logicality, at least, of the first similar closure of high-temperature hexagonal structures was indicated repeatedly by Khan in the 18 studies related to the R–Co systems. In particular, he wrote: “It should be noted that SmCo_5+X_ and Sm_2_Co_17_(h) are isostructural (i.e., of the TbCu7-type with disordered substitutions). It means that we should obtain a homogeneous region from SmCo_5+X_ to Sm_2_Co_17_(h) in the phase diagram, at least, at temperatures near to the solidus. What we experimentally obtain is a eutectic in the neighborhood of SmCo_7_. An explanation for this discrepancy is still being sought” [34].

Thus, at high temperatures, the 1:7-based solid solution range in the quasi-ternary phase diagram is present in the form of an asymmetric biconvex (toward the decreasing temperature) lens adjacent to the high-temperature homogeneity ranges of the 2:17 and 2:7 phases in their wider ranges.

It should be noted that the given sketches of the phase diagrams are related to certain relations of 3d elements, which correspond to the formula R_1-X_Zr_X_(Co_0.70_Cu_0.09_Fe_0.21_)_Z_.

## 6. Conclusions

We present here an original view on the issues of the structural formation of the (R,Zr)(Co,Cu,Fe)_Z_ alloys in the concentration ranges promising for the production of high-coercivity permanent magnets. These findings were based on a quantitative analysis of the microstructures of the first, most crude level of heterogeneity of these alloys in a highly coercive state.

It was shown that:The formula of the alloys R_1-X_Zr_X_(Co,Cu,Fe)_Z_ is physically most reasonable from the viewpoint of the effect of main elements with the qualitatively different electron shells (4f–R, 4d–Zr and 3d–Co,Cu,Fe) on the structure and properties of the studied alloys, and with allowance for the fact that the initial structural state is the matrix being the single-phase disordered 1:7H solid solution, and almost all intermediate and final phases (except the 2:17R phase) formed in the course of phase transformations corresponded better to the formula R_1-X_Zr_X_(Co,Cu,Fe)_Z_.The microstructure of the R_1-X_Zr_X_(Co,Cu,Fe)_Z_ alloys in optical resolutions was formed by three structural components based on the 1:5 (A), 2:17 (B), and 2:7 (C) phases; the qualitative relationships A:B:C varied within wide ranges as the chemical composition changed monotonously.For the studied composition ranges of the Sm_1-X_Zr_X_(Co,Cu,Fe)_Z_ alloys of all experimental series, the increase in the relative content of 3d elements (z) was accompanied by monotonic changing of the volumes of structural components A and B, corresponding to 90% of the total alloy volume from the dominant volume of structural component A to the dominant volume of structural component B.The hysteretic properties of the Sm_1-X_Zr_X_(Co,Cu,Fe)_Z_ alloys in the high-coercivity state were strictly controlled by the volume percentage ratio of the structural components A and B.The dominant volume of the structural component A in the alloy ensured the ultimate squareness of the hysteresis loop of the samples; the equality of the volumes V_A_ = V_B_ corresponded to the combination of the high coercive force (H_CJ_) and ultimate squareness of the hysteresis loop; the predominance of the volume of component B was accompanied by the decrease in the squareness of the hysteresis loop.The increase in the relative Zr content (x) in the Sm_1-X_Zr_X_(Co,Cu,Fe)_Z_ alloys in the high-coercivity state was accompanied by the decrease in the precision of the dependences of the coercive force on z and the shift of alloy compositions with the equal volumes V_A_ = V_B_ toward the lower relative content of 3d-elements (z).At temperatures corresponding to the supersaturated solid solution treatment, the section of the ternary diagram of the Co–Sm–Zr system undergoes a significant transformation upon introduction of Cu and Fe into the alloy and the transition to the (Co,Cu,Fe)–Sm–Zr quasi-ternary section.In the quasi-ternary phase diagram of (Co,Cu,Fe)–Sm–Zr, at SSHT temperatures, the region of the 1:7-phase-based solid solution had the form of an asymmetric lens, biconvex toward lower temperatures; the edges of the lens were adjacent to the homogeneity ranges of the 2:17 and 2:7 phases in their widest, high-temperature regions.

In future studies, we will consider the peculiarities of magnetization reversal and the composition of the structural components of the (Sm,Zr)(Co,Cu,Fe)_Z_ alloys in high-coercivity state for composition ranges that are of importance in the manufacture of permanent magnets.

## Figures and Tables

**Figure 1 materials-13-03893-f001:**
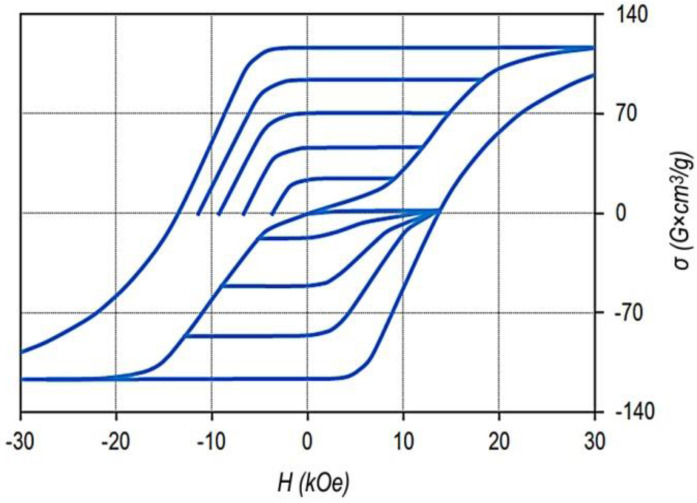
Major and minor hysteresis loops of pseudo-single crystal Sm_0.87_Zr_0.13_(Co_0.60_Cu_0.070_Fe_0.240_)_6.5_ sample in the high-coercivity state; (bottom) and (top) curves correspond to the magnetization curves of the sample demagnetized by reverse field and alternating filed, respectively (VSM, without allowance for the demagnetizing factor, N = 1/3).

**Figure 2 materials-13-03893-f002:**
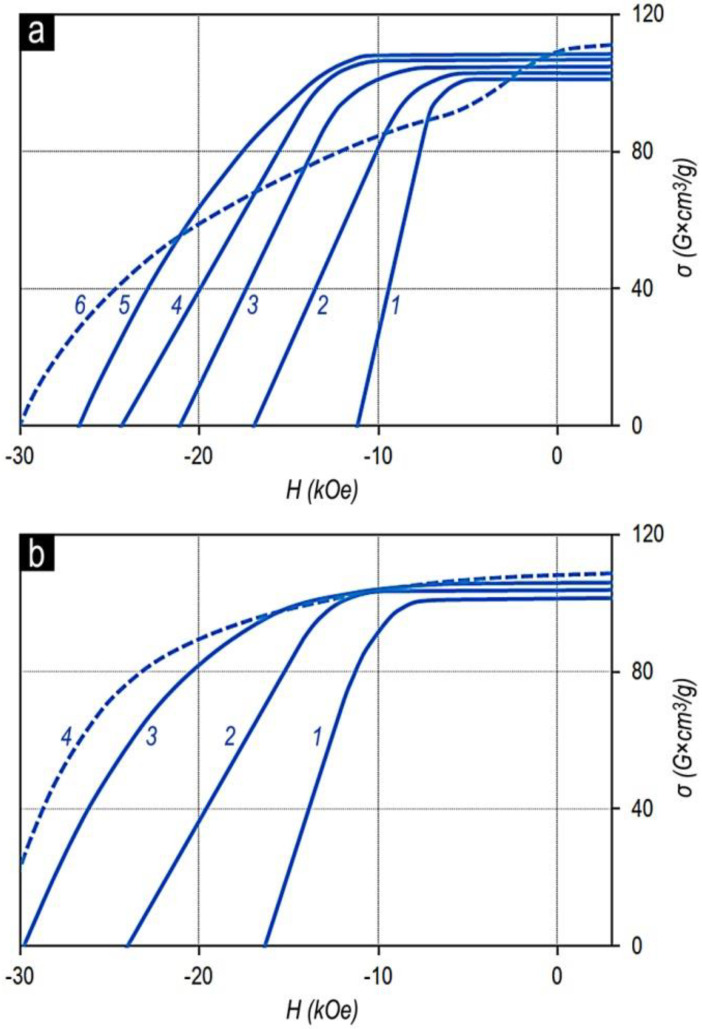
Magnetization reversal portions of major hysteresis loops of pseudo-single crystal samples in high-coercivity state: (**a**) Sm_0.85_Zr_0.15_(Co_0.702_Cu_0.088_Fe_0.210_)*z* with z = 6.0 (1); 6.2 (2); 6.3 (3); 6.4 (4); 6.5 (5) and 6.8 (6), and (**b**) Sm_1-X_Zr_X_(Co_0.702_Cu_0.088_Fe_0.210_)_6.4_ with х = 0.13 (1); 0.15 (2); 0.17 (3) and 0.19 (VSM, without allowance for the demagnetizing factor, N = 1/3).

**Figure 3 materials-13-03893-f003:**
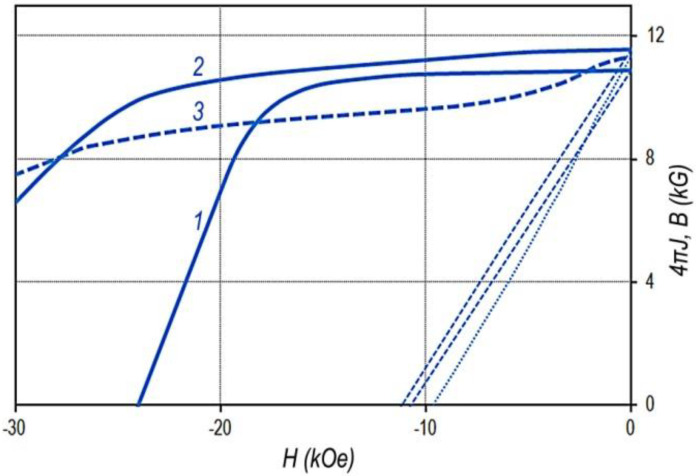
Magnetization reversal portions of hysteresis loops of sintered powder samples in the high-coercivity state for the Sm_0.85_Zr_0.15_(Co_0.702_Cu_0.088_Fe_0.210_)_z_ alloys with z = 6.0 (1); 6.4 (2); 6.7 (3).

**Figure 4 materials-13-03893-f004:**
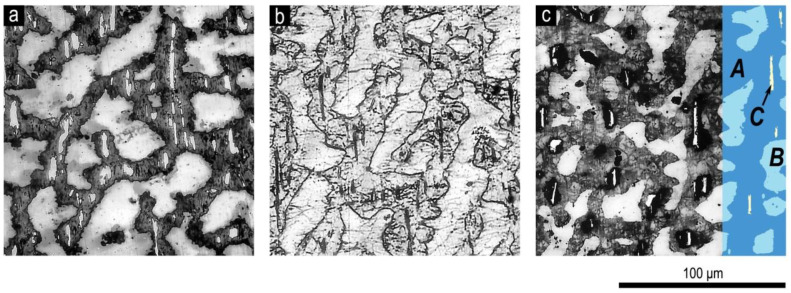
Microstructure on the prismatic plane of the pseudo-single crystal samples of the Sm_0.85_Zr_0.15_(Co_0.702_Cu_0.088_Fe_0.210_)_6.4_ alloy in the (**a**) as-cast state, (**b**) after solid-solution heat treatment, and (**c**) after complete cycle of heat treatment (optical microscope, etching with 5% alcohol solution of HNO_3_). The easy magnetization axis is in the image plane and strictly horizontal. Definitions for structural components for A, B and C in (**c**) are given in the text.

**Figure 5 materials-13-03893-f005:**
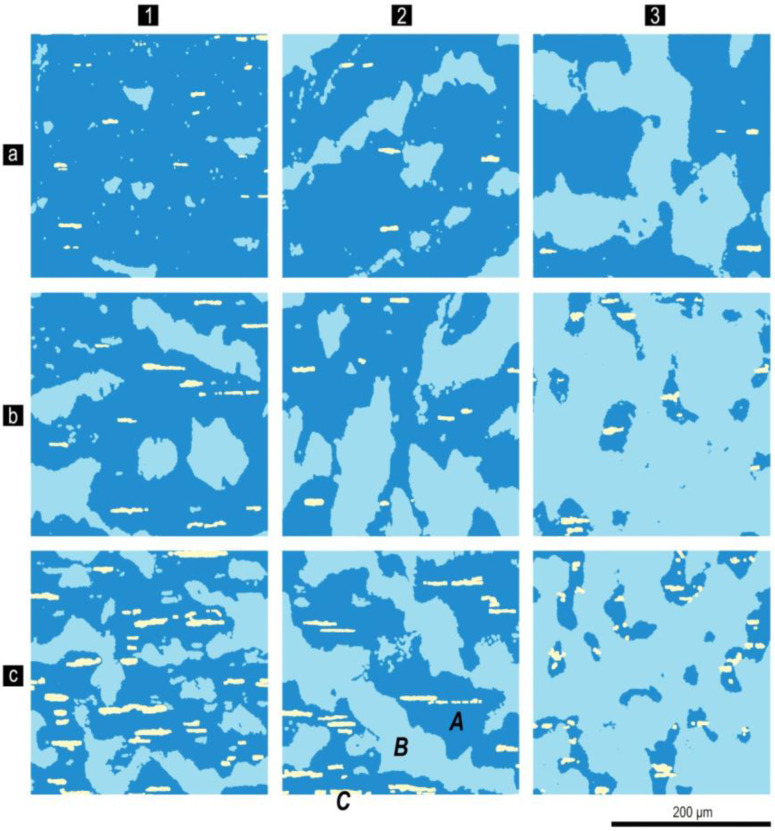
Changes of relationships of structural phase components A, B, and C (are shown in panel c and in Figure 4) in pseudo-single crystal samples of Sm_1-X_Zr_X_(Co_0.702_Cu_0.088_Fe_0.210_)_z_ alloys in the high-coercivity state with x = 0.13 (**a**1—3); 0.15 (**b**1—3) and 0.19 (**c**1—3), and z = 6.2 (1); 6.4 (2); 6.7 (3) (prismatic plane, the easy magnetization axis is in the image plane and strictly vertical).

**Figure 6 materials-13-03893-f006:**
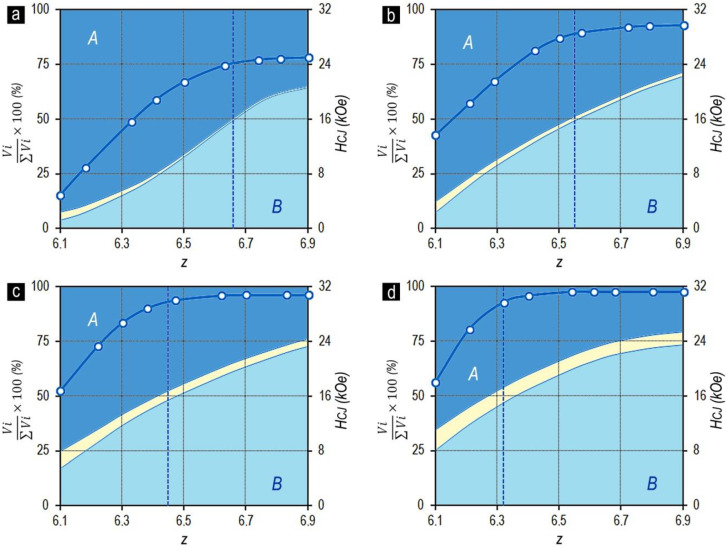
Dependences of the volume percentage of structural components A, B, and C ((V_i_/ΣV_i_) × 100) and coercive force (H_CJ_) of the Sm_1-X_Zr_X_(Co_0.702_Cu_0.088_Fe_0.210_)_z_ alloys with x = 0.13 (**a**), 0.15 (**b**), 0.17 (**c**), and 0.19 (**d**) on the chemical composition z. The vertical dashed line shows the compositions with equal volume percentage of structural components A and B.

**Figure 7 materials-13-03893-f007:**
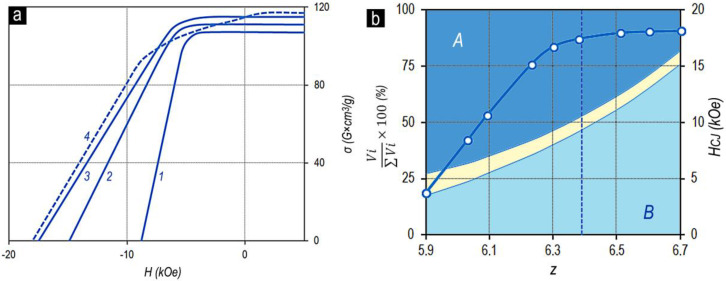
(**a**) Magnetization reversal portions of major hysteresis loops and (**b**) dependences of the volume percentage of structural phase components A, B, and C ((V_i_/ΣV_i_)x100) and coercive force (H_CJ_) of the Sm_0.85_Zr_0.15_(Co_0.665_Cu_0.075_Fe_0.260_)z alloys with z = 6.0(1); 6.2(2); 6.4(3) and 6.5(4) on the chemical composition z (VSM, without allowance for the demagnetizing factor, N = 1/3).

**Figure 8 materials-13-03893-f008:**
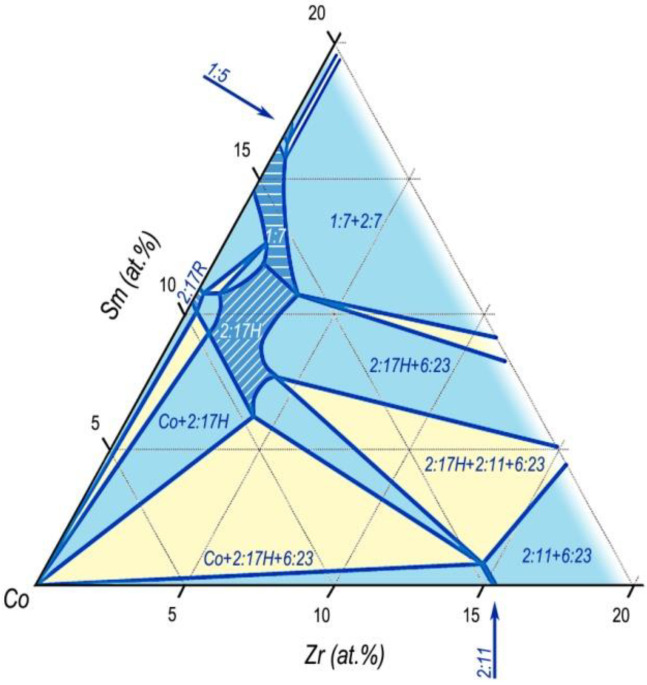
Isothermal section of Co corner of the ternary Co–Sm–Zr phase diagram at 1200 °C (compilation of data of Nishio et al. [25]).

**Figure 9 materials-13-03893-f009:**
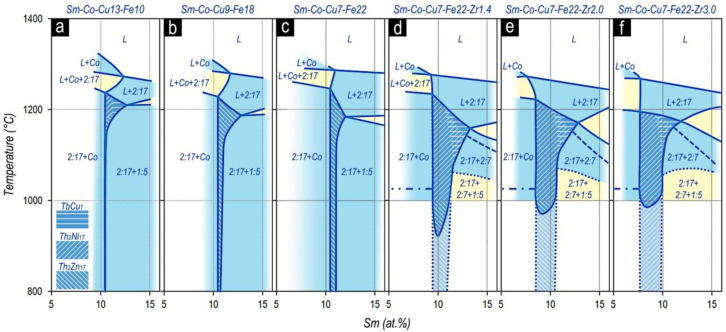
Phase formation in the (Sm,Zr)(Co,Cu,Fe)_Z_ alloys was demonstrated in partial isopleths of quasi-ternary phase diagrams (**a**–**f**) in the homogeneity range of the 2:17 phase (compilation based on data of Morita et al. [22,23,24] with some corrections).

**Figure 10 materials-13-03893-f010:**
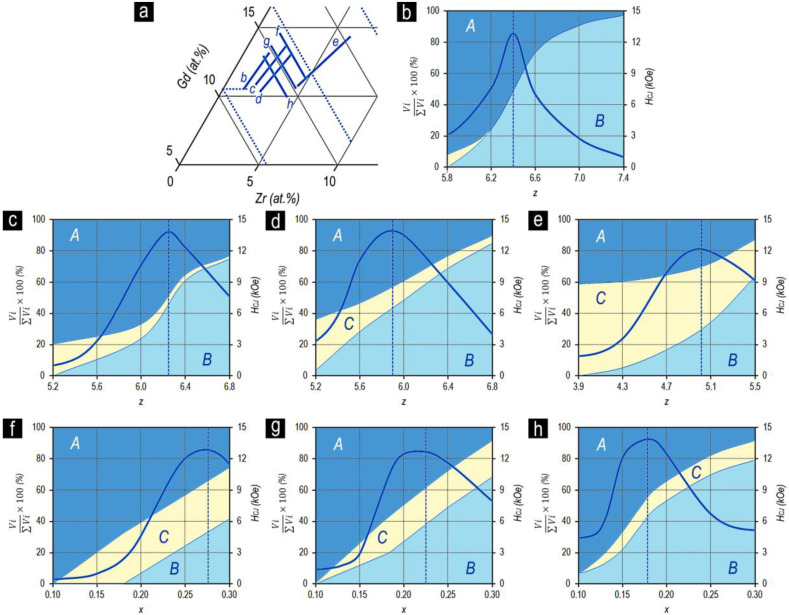
(**a**) Experimental series of Gd_1-X_Zr_X_(Co_0.70_Cu_0.09_Fe_0.21_)z alloys in the section of the cobalt corner of the quasi-ternary (Co_0.70_Cu_0.09_Fe_0.21_)–Gd–Zr phase diagram and dependences of the volume percentage of structural components A, B, and C ((Vi/ΣVi) × 100) and coercive force (H_CJ_) on the chemical composition z for the alloys with x = 0.11 (**b**), 0.15 (**c**), 0.19 (**d**), and 0.30 (**e**) and on x for fixed z = 5.2 (**f**), 5.6 (**g**), and 6.0 (**h**). Based on the compilation of results available in Lyakhova et al. [11,12].

**Figure 11 materials-13-03893-f011:**
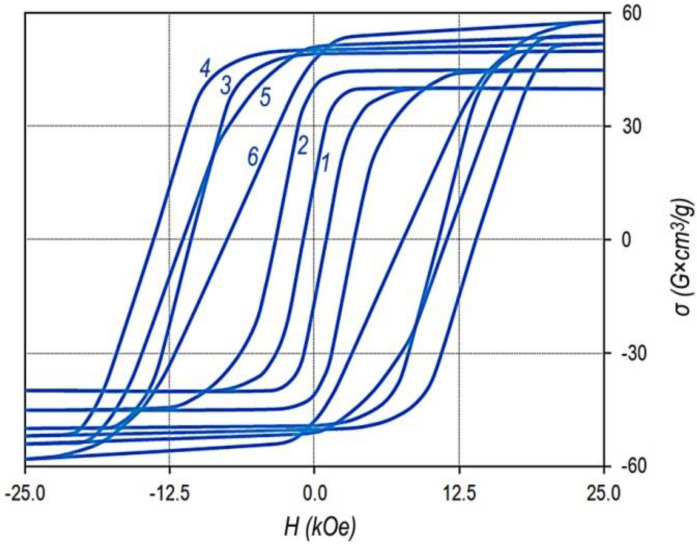
Major hysteresis loops of pseudo-single crystal samples of Gd_0.85_Zr_0.15_(Co_0.70_Cu_0.09_Fe_0.21_)_Z_ alloys with z = 5.2(1), 5.6(2), 6.0(3), 6.2(4), 6.4(5), and 6.8(6); VSM, without allowance for the demagnetizing factor, N = 1/3); compilation of data of Lyakhova et al. [11].

**Figure 12 materials-13-03893-f012:**
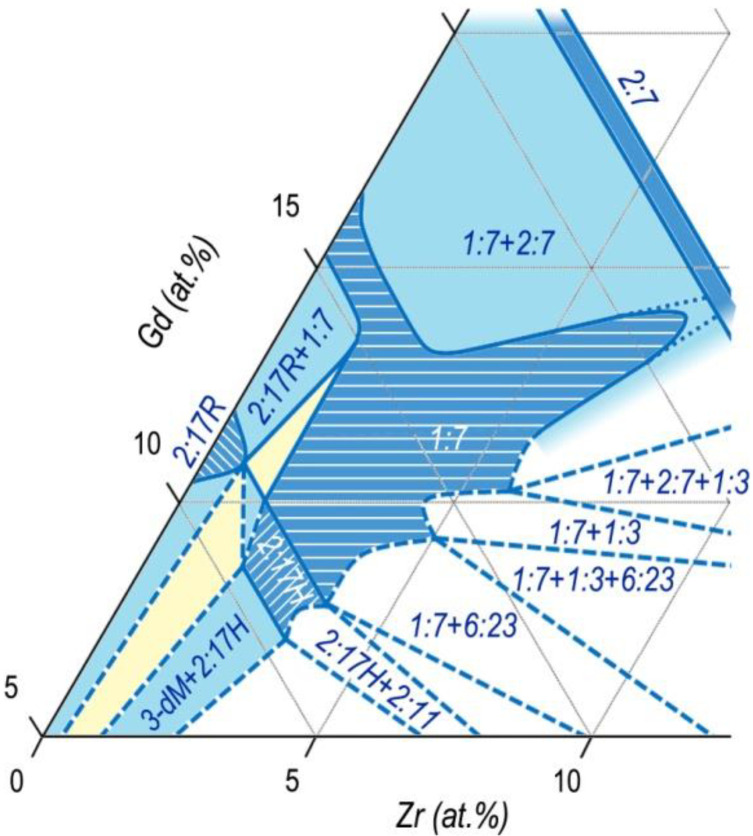
Sketch of the 3-dM angle of the quasi-ternary (Co,Cu,Fe)–Gd–Zr phase diagram for a temperature range of 1160–1190 °C.

**Figure 13 materials-13-03893-f013:**
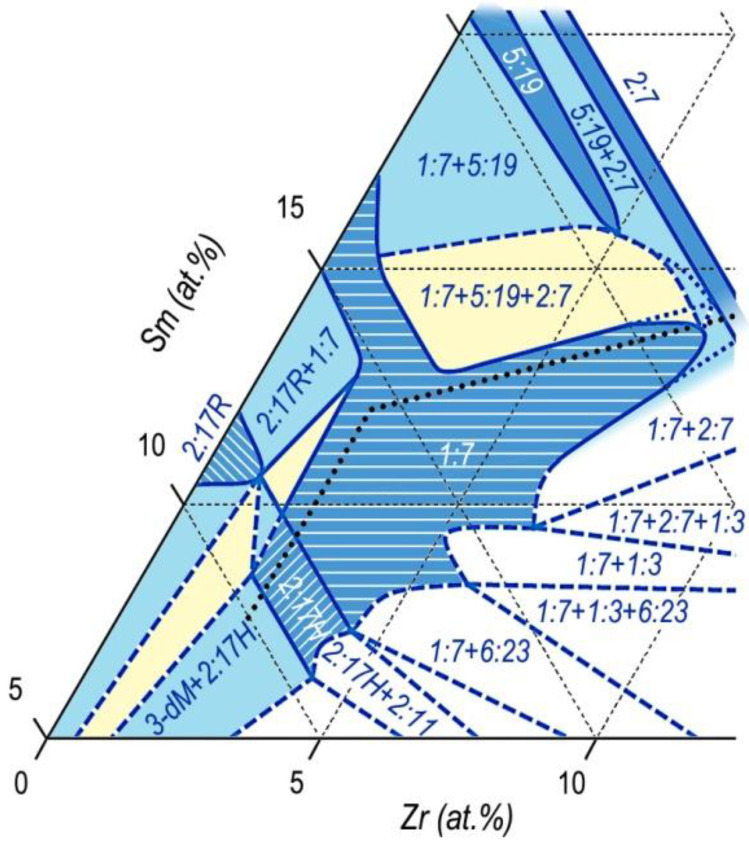
Sketch of the 3-dM angle of the quasi-ternary (Co,Cu,Fe)–Sm–Zr phase diagram for a temperature range of 1160–1190 °C.

**Figure 14 materials-13-03893-f014:**
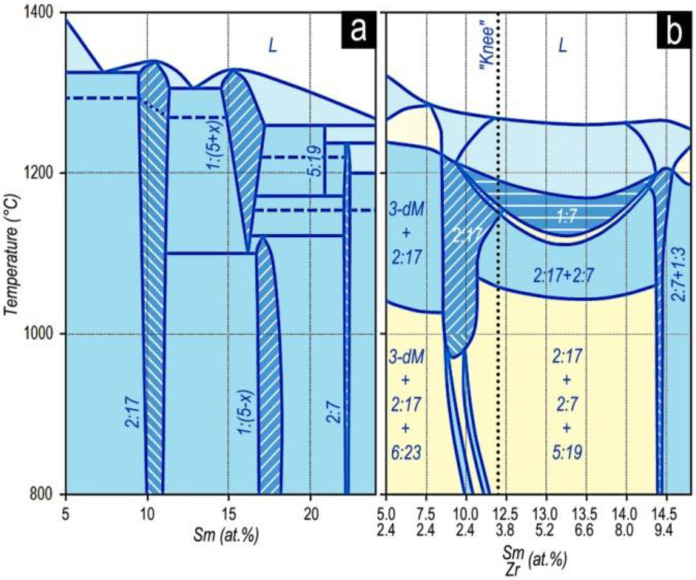
(**a**) Portion of the binary Co–Sm phase diagram (compilation of Massalski et al. [33]) for the actual composition range and (**b**) its transformation with adding Zr, Cu, and Fe into the isopleth of the (Co,Cu,Fe)–Sm–Zr diagram (dashed line in Figure 13).

**Table 1 materials-13-03893-t001:** Chemical composition of the experimental alloys Sm_1-X_Zr_X_(Co_1-a-b_Cu_a_Fe_b_)_Z_.

Series no.	x	a	b	z
**1**	0.13	0.088	0.210	6.0–6.8
**2**	0.15	0.088	0.210	6.0–6.8
**3**	0.17	0.088	0.210	6.0–6.8
**4**	0.19	0.088	0.210	6.0–6.8
**5**	0.15	0.075	0.260	6.0–6.8
**6**	0.13	0.070	0.240	6.1–6.5

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
