# Peer review of "Structure of Alloys for (Sm,Zr)(Co,Cu,Fe)Z Permanent Magnets: First Level of Heterogeneity"

_materials, 2020, doi:10.3390/ma13173893_

Round 1

Reviewer 1 Report

Thanks to the authors, to work on the exciting and a challenging subject. The data presented is supported by the research on the area done by other research groups. 

  • In the result section authors mentioned about the EDX result and composition corresponds to that. Can authors couple EDX data in a figure and which structure they correspond to make the manuscript complete?
  • Figure 8, 9, 10, 11, 14 are compilation of previous research work, did authors get permission from the original authors?
  • Figure 9 is a compilation of data with some correction from Morita et al. work. What are the correction authors refer to?

There is a minor change needed on the manuscript.

  1. Line 35 - keep only [7] instead of [for example, 7, etc.]

Author Response

We thank the reviewer for the consideration of our paper.

We checked the figures, delete references from figure captions and explain the following.

  1. All illustrative information in our paper is given logically. This certain logic is outlined in the beginning of section 5. Discussion.

Note. It is difficult to compared data on the phase diagrams constructed by different investigators, which use different coordinate systems. Because of this, we compile the available data and construct the ternary phase diagrams in the form more convenient for the present study, namely, cobalt always is in the left corner of binary, quasi-binary sections and ternary phase diagrams; the rare-earth metal is always in the top corner of the ternary systems. The alloying component is in the right corner.”

  1. None of the original images of other authors are given in our paper. We present digitized and corrected data of other authors.
  1. We assemble and arrange data of different authors to follow the logic of presentation.
  1. When plotting our images, we correct original data that are used.
  1. In this case, we without fail give references for data used in figures.

None of our figures repeat authors’ figures.

Per se, we give convenient references on the opinion of one or another author and present our interpretation.

Specially;

Fig. 8 Fig. 8. Isothermal section of Co corner of the ternary Co-Sm-Zr phase diagram at 1200 °Ð¡ (compilation of data of Nishio et al. [24]).

Figure 8 was redrawn so that Co corner is shown on the left, as is required by the logic of the presentation of the material of our article. 

Fig. 9. Phase formation in the (Sm,Zr)(Co,Cu,Fe)z alloys demonstrated in partial isopleths of quasi-ternary phase diagrams in the homogeneity range of the 2:17 phase (compilation based on data of Morita et al. [21-23] with some corrections).

Data given in this figure compile data from six figures given in three papers. Our figures

  1. i) as the paper logic demands, data are given in horizontal position as compared to the original data
  1. ii) data are given in the form convenient for understanding, without gaps for the discussed temperature range of phase transformations.

iii) figures differ substantially from original ones. We correct phase equilibria in accordance with fundamental canons and modern knowledge on the discussed system

Fig. 10. (a) Experimental series of Gd1-XZrX(Co0.70Cu0.09Fe0.21)Z alloys in the section of cobalt corner of the quasi-ternary (Co0.70Cu0.09Fe0.21)-Gd-Zr phase diagram and dependences of the volume percentage of structural components A, B, and C ((Vi/ΣVi)×100) and coercive force (HCJ) on the chemical composition z for the alloys with x=0.11 (b), 0.15 (c), 0.19 (d), and 0.30 (e) and on x for fixed z = 5.2 (f), 5.6 (g), and 6.0 (h) (Based on compilation of results available in Lyakhova et al. [10, 11]). 

Fig. 10 is compilation of data of [10, 11]) given in another manner, and some own considerations were added.

Fig. 11. Major hysteresis loops of pseudo-single crystal samples of Gd0.85Zr0.15(Co0.70Cu0.09Fe0.21)Z alloys with z = 5.2(1), 5.6(2), 6.0(3), 6.2(4), 6.4(5), and 6.8(6); (VSM, without allowance for the demagnetizing factor, N = 1/3); compilation of data of Lyakhova et al. [10]).

Fig. 11. Data for three of six samples in the hysteresis loops were corrected and the data was replotted.

Fig. 14. (a) Portion of the binary Co-Sm phase diagram (compilation of Massalski et al. [32]) for the actual composition range and (b) its transformation with adding Zr, Cu, and Fe into the isopleth of the Co,Cu,Fe)-Sm-Zr diagram (dashed line in Fig. 13).

Fig. 14. The left image is taken from handbook “T.B. Massalski, J.L. Murray, L.H. Benett, H. Baker, Binary alloy Phase Diagrams, American Society for Metals, 1990” (which can be shown without copyright issue). This portion is given for readers for convenience. The right portion is completely our original data (which were not published earlier).

So, we assume that no copyright issue should be.

English was checked carefully in the text.

  1. Line 35 - keep only [7] instead of [for example, 7, et]

The correction was made in the text (shown by yellow).

Reviewer 2 Report

This work presents a study of structure formation of (R,Zr)(Co,Cu,Fe)z alloy dedicated to manufacturing permanent magnets development. The structures of these alloys are characterized by optical and magnetic measurements. Results are compared and analyzed with other results from the literature.

This paper is well written, and the study of heterogeneity is rich in information. The analysis is very interesting. Compilation of available data and construction of the ternary phase diagrams in the form convenient for this study brings excellent value to this work.

As you can see above, the manuscript could be performed on a further point that has been noted.

  • More citations about the interest of these alloys in the manufacturing of permanent magnet would be interesting.
  • Some information about the magnetic characterization has to be added.
  • Some minor corrections are needed

p.1 l. 35:  citations have to be developed.

p.1 l.41: the importance of this study for the manufacturing process has to be developed. Citations, as well as numerical quantifications of the sensibility of permanent magnets properties to interrelations of the chemical and phase compositions, are necessary to explain the real interest of this work.

p.2 l.64: motivation of this assumption has to be developed.

p.3 l.101-103: citations are required, to describe experimental results and the modern experimental equipment used.

p.3. l.106: the use of the terms “impressive” and “large information” are inappropriate. Developpement about the capacity of the considered setup has to be detailed.

p.4 l.148: citations required to define the heat treatment conditions considered.

  1. 4 l.156: the specific facility used has to be described

p.4 l.159: the model of the VSM has to be detailed. Indications about the accuracy of measurement would be interesting.

p.4 l.160: how this magnetic field of 100kOe (far beyond the maximum value of the VSM) has been applied? Details are needed.

p.4 l.161: σS has to be defined

p.4 l.164: what is the frequency of the demagnetization process used? Indications about the accuracy of the demagnetized state reached (remanent magnetic induction after demagnetization process, for example) would be interesting.

p.4 l.167: the specific facility has to be detailed

p.4 l.169: citations needed to describe standard techniques

p.5 l.188: the dimensions of sintered samples have to be detailed. A brief description of the measurement setup has to be added.

The term “field” is confused with "filed" on some sentences

Fig 1 is uncleared. It would be clearer to present the major loops using a different color and to separate the bottom and the top curves to show complete loops in each case. This presentation is confusing. Demagnetizing factor has to be defined. The sentence "VSM, without allowance 200 for the demagnetizing factor, N = 1/3” is unclear too.

Some information about the excitation signal are missing: frequency, form, control.

p.5 l.214: J has to be defined. The relation and constant used to access J have to be detailed. In the same way, (BH)max has to be defined, and the calculation process developed.

Figure 4: Elements A, B, C have to be defined in the caption of the figure.

p.7 l.263: reference to figure 4c has to be added.

Figure 5: it could be useful to precise the definition of A B C or to refer to the definition of Figure 4. Notation panel a b c seems to be confusing with the SPC A B C.

Author Response

We thank the reviewer for the consideration of our paper.

English was checked carefully in the text.

The following corrections were made and marked by yellow in the text.

p.1 l. 35:  citations have to be developed

Was

“There are sufficiently exotic variants of the phase structure at boundary-cell interface [for example, 7, etc.] in the literature.”

Improved variant

There are sufficiently exotic variants of the phase structure at boundary-cell interface. For example, Popov et. al [7] assume that the boundary phase is separated so that Cu is localized in the 1:5 phase near the (1:5)/(2:17) interface rather than in the center of the 1:5 boundary phase. However, the structural peculiarity of the boundary is the alternation of phase layers strictly across the “c” axis, which is common for the whole anisotropic massive of samples [3, 8]. Within such a structure, the formation of additional phase separation along the most low-size coordinate of boundary structural element is not likely energetically reasonable. Moreover, this assumption certainly contradicts the fact that the Cu concentration in the center of boundary phase in ternary junctions of cells is maximal [8].

p.1 l.41: the importance of this study for the manufacturing process has to be developed. Citations, as well as numerical quantifications of the sensibility of permanent magnets properties to interrelations of the chemical and phase compositions, are necessary to explain the real interest of this work.

Was

“The aim of the present study is detailing the interrelations of the chemical and phase compositions of the (Sm,Zr)(Co,Cu,Fe)z alloys in the composition ranges, which are of importance for manufacturing processes of permanent magnets”.

Imtroved

“The aim of the present study is detailing the interrelations of the chemical and phase compositions of the (Sm,Zr)(Co,Cu,Fe)z alloys in the composition ranges, which are of importance for manufacturing processes of permanent magnets because of the functionality of their phase structure. The magnetic hardness of the (Sm,Zr)(Co,Cu,Fe)z alloy for permanent magnets is ensured by its precipitation hardening in the course of complex heat treatment”

p.2 l.64: motivation of this assumption has to be developed.

We think that no corrections should be made because this is not assumption; this is the statement of existing situation. All previously reported dependences properties vs composition for the Sm-Zr-Co-Cu-Fe alloys are constructed taking into account formulas Sm2(Co,Fe,Cu,Zr)17 and Sm(Co,Cu,Fe,Zr)z. We believe that the use of the formula (Sm,Zr)(Co,Cu,Fe)z is more physically reasonable. We prove this by the data reported in the present paper.

p.3 l.101-103: citations are required, to describe experimental results and the modern experimental equipment used.

The references [3, 5-8] were added.

“There are many results of experimental investigations of (Sm,Zr)(Co,Cu,Fe)z materials for permanent magnets in the literature, which were obtained by modern experimental equipment that is perfect from the point of view methodological approach [3, 5-8].”

The chemical analysis of alloys was performed by optical emission spectroscopy with inductively coupled plasma (ULTIMA 2 Jobin-yvon ICP-OES).

p.3. l.106: the use of the terms “impressive” and “large information” are inappropriate. Developpement about the capacity of the considered setup has to be detailed.

Was

“Results obtained by high-resolution transmission electron microscopy (HRTEM), nanoprobe energy dispersive X-ray  spectroscopy (EDXS), nano-beam diffraction (NBD), etc. methods are impressive and contain a large body of information about the structure of samples of sintered (Sm,Zr)(Co,Cu,Fe)z magnets after different stages of thermal aging”.

Improved

“Results obtained by high-resolution transmission electron microscopy (HRTEM), nanoprobe energy dispersive X-ray spectroscopy (EDXS), nano-beam diffraction (NBD), etc. methods allowed one to estimate the fine structure of (Sm,Zr)(Co,Cu,Fe)z sintered magnet samples using results of local analysis of some phases [3, 5, 6, 8]”

p.4 l.148: citations required to define the heat treatment conditions considered.

Impoved

The heat treatment conditions are similar to those described in the literature [2-8].

  1. 4 l.156: the specific facility used has to be described

Was

“Pseudo-single crystal samples of each of the compositions in as-cast and heat-treated (under different conditions) states were ground to form balls 2.5-3.5 mm in diameter using a specific facility”.

Improved

“Pseudo-single crystal samples of each of the compositions in as-cast and heat-treated (under different conditions) states were ground to form balls 2.5-3.5 mm in diameter by grinding with gaseous nitrogen supplied under a pressure between two abrasive caps.”

p.4 l.159: the model of the VSM has to be detailed. Indications about the accuracy of measurement would be interesting.

Corrected

The magnetic measurements of major and minor hysteresis loops of pseudo-single crystal spherical samples were performed at room temperature using a vibrating sample magnetometer (VSM LDJ Electronics Inc., Model 9600) (HMAX = 30 kOe)).

p.4 l.160: how this magnetic field of 100kOe (far beyond the maximum value of the VSM) has been applied? Details are needed.

Corrected

To measure the major hysteresis loop, the samples were preliminarily magnetized in a magnetic field pulsed magnetization unit (Ningbo Canmag Electronics Co., Model KCJ-3560G) of no less than 100 kOe.

p.4 l.161: σS has to be defined

Corrected

The VSM was graduated using a Ni standard with specific magnetization σSNi = 54.4 G×cm3/g.

p.4 l.164: what is the frequency of the demagnetization process used? Indications about the accuracy of the demagnetized state reached (remanent magnetic induction after demagnetization process, for example) would be interesting.

We use quasistatic field. Because of this, no frequency can be considered. A sample is demagnetized to “0” (taking into account the sensitivity of VSM that is 1×10-5 emu). In this case, we can study the sample with SEM.

p.4 l.167: the specific facility has to be detailed

Corrected sentence

“The samples completely evaluated with respect to magnetic parameters were demagnetized using alternating magnetic field of VSM with decreasing amplitude and were fixed in mandrels with a fast curing epoxy compound.”

p.4 l.169: citations needed to describe standard techniques

Corrected sentence

“The sections were prepared in accordance with standard techniques using diamond pastes with a gradual reduction in grain size.”

p.5 l.188: the dimensions of sintered samples have to be detailed. A brief description of the measurement setup has to be added.

Corrected sentence

“The magnetic properties of sintered samples (Ø15×7 mm) were measured in fields of to 30 kOe using a completely closed magnetic circuit and an automatic recording flux meter (B-H tracer, LDJ Electronics Inc., Model 5500H)”

English was checked over the text.

Fig 1 is uncleared. It would be clearer to present the major loops using a different color and to separate the bottom and the top curves to show complete loops in each case. This presentation is confusing. Demagnetizing factor has to be defined. The sentence "VSM, without allowance 200 for the demagnetizing factor, N = 1/3” is unclear too.

Some information about the excitation signal are missing: frequency, form, control.

Information in Fig. 1 is given in traditional representation that is taken in the majority of publications related to permanent magnets. Measurements with a VSM are standard and frequency, form, control of excitation signal are beyond the matter of the paper.

The major loop envelopes all minor loops.

The demagnetizing factor of a spherical sample in the open magnetic circuit of VSM is N=1/3. As a rule, results of measurements with VSM are given in the “as-obtained” form “(VSM, without allowance for the demagnetizing factor, N = 1/3)”

p.5 l.214: J has to be defined. The relation and constant used to access J have to be detailed. In the same way, (BH)max has to be defined, and the calculation process developed.

J is the magnetic moment of М of a unity volume of a body J=M/V. 4πJS и (BH)MAX are standard parameters that are the saturation magnetization and maximum energy product of hard magnetic materials, respectively. The parameters are international and generally accepted to characterize the permanent magnets.

Figure 4: Elements A, B, C have to be defined in the caption of the figure.

p.7 l.263: reference to figure 4c has to be added.

Corrected caption for Fig. 4

Figure 4. Microstructure on the prismatic plane of the pseudo-single crystal samples of the Sm0.85Zr0.15(Co0.702Cu0.088Fe0.210)6.4 alloy in the (a) as-cast state, (b) after solid-solution heat treatment, and (c) after complete cycle of heat treatment (optical microscope, etching with 5 % alcohol solution of HNO3). The easy magnetization axis is in the image plane and strictly horizontal. Definitions for structural components A, B and C in (c) are given in the text.

Figure 5: it could be useful to precise the definition of A B C or to refer to the definition of Figure 4. Notation panel a b c seems to be confusing with the SPC A B C.

Over the text, structural phase components (SPC) are given with capital letters (in contrast to panels a b c). Reference to Fig. 4 is added in Fig. 5.

Figure 5. Changes of relationships of structural phase components A, B, and C (are shown in panel C2 and in Fig. 4) in pseudo-single crystal samples of Sm1-XZrX(Co0.702Cu0.088Fe0.210)z alloys in the high-coercivity state with x = 0.13 (a 1—3); 0.15 (b 1—3) and 0.19 (c 1—3), and z = 6.2 (1); 6.4 (2); 6.7 (3) (prismatic plane, the easy magnetization axis is in the image plane and strictly vertical).

Reviewer 3 Report

The paper seems to be interesting and can be accepted after addressing the below mentioned comments.

  1. The English needs to be revised throughout the manuscript.
  2. The paragraphing should be reconsidered.
  3. Rewrite introduction by adding more up-to-date references and highlight the problems in the previous works.
  4. The abstract need to be revised as per the guidelines.
  5. The article motivation is rather weak and unclear.
  6. Compare this research’s findings with state of the art and explain the benefits of this approach.
  7. Conclusion should be revised based on some scientific explanation.

Author Response

We thank the reviewer for the consideration of our paper. However, the comments are so that it is difficult to perform revision of the paper.

  1. The English needs to be revised throughout the manuscript

English was checked carefully.

  1. The paragraphing should be reconsidered.

The paragraphing corresponds to the logic of presentation and is concrete.

  1. Rewrite introduction by adding more up-to-date references and highlight the problems in the previous works.

We used references to both traditional and modern knowledge about the permanent magnets of the Sm(Gd)-Co-Zr-Cu-Fe systems.

  1. The abstract need to be revised as per the guidelines.

Abstract corresponds to the considered problem.

  1. The article motivation is rather weak and unclear.

The motivation of the article is

Lines 36-39

“It should be recognized that currently there is no objective holistic view of the formation of high-coercivity structure of the (Sm,Zr)(Co,Cu,Fe)z alloys. This hinders the progress in the development and improvement of both new compositions of alloys for permanent magnets and manufacturing processes for them.”

  1. Compare this research’s findings with state of the art and explain the benefits of this approach.

We represent an original approach and all obtained and considered results are compared with available literature data.

  1. Conclusion should be revised based on some scientific explanation.

The conclusions are constructed in accordance with the aim of the paper.

“The aim of the present study is detailing the interrelations of the chemical and phase compositions of the (Sm,Zr)(Co,Cu,Fe)z alloys in the composition ranges, which are of importance for manufacturing processes of permanent magnets because of the functionality of their phase structure. The magnetic hardness of the (Sm,Zr)(Co,Cu,Fe)z alloy for permanent magnets is ensured by its precipitation hardening in the course of complex heat treatment. The study is also aimed at concretizing the phase transformations of the Sm-Zr-Co-Cu-Fe system in accordance with our understanding of occurred processes and phenomena”